# A Dipeptidyl Peptidase-4 Inhibitor Inhibits Foam Cell Formation of Macrophages in Type 1 Diabetes via Suppression of CD36 and ACAT-1 Expression

**DOI:** 10.3390/ijms21134811

**Published:** 2020-07-07

**Authors:** Michishige Terasaki, Hironori Yashima, Yusaku Mori, Tomomi Saito, Takanori Matsui, Munenori Hiromura, Hideki Kushima, Naoya Osaka, Makoto Ohara, Tomoyasu Fukui, Tsutomu Hirano, Sho-ichi Yamagishi

**Affiliations:** 1Department of Medicine, Division of Diabetes, Metabolism, and Endocrinology, Showa University School of Medicine, Tokyo 142-8666, Japan; yashima@med.showa-u.ac.jp (H.Y.); u-mori@med.showa-u.ac.jp (Y.M.); saito_to@cnt.showa-u.ac.jp (T.S.); hiromura@med.showa-u.ac.jp (M.H.); hkushima@med.showa-u.ac.jp (H.K.); n.oosaka0709@gmail.com (N.O.); s6018@nms.ac.jp (M.O.); showauft@med.showa-u.ac.jp (T.F.); shoichi@med.showa-u.ac.jp (S.Y.); 2Department of Pathophysiology and Therapeutics of Diabetic Vascular Complications, Kurume University School of Medicine, Kurume 830-0011, Japan; matsui_takanori@med.kurume-u.ac.jp; 3Diabetes Center, Ebina General Hospital, Ebina 243-0433, Japan; hirano@med.showa-u.ac.jp

**Keywords:** DPP-4 inhibitors, macrophage foam cell formation, CD36, ACAT-1, AGEs, type 1 diabetes

## Abstract

Dipeptidyl peptidase-4 (DPP-4) inhibitors have been reported to play a protective role against atherosclerosis in both animal models and patients with type 2 diabetes (T2D). However, since T2D is associated with dyslipidemia, hypertension and insulin resistance, part of which are ameliorated by DPP-4 inhibitors, it remains unclear whether DPP-4 inhibitors could have anti-atherosclerotic properties directly by attenuating the harmful effects of hyperglycemia. Therefore, we examined whether a DPP-4 inhibitor, teneligliptin, could suppress oxidized low-density lipoprotein (ox-LDL) uptake, foam cell formation, *CD36* and acyl-coenzyme A: cholesterol acyltransferase-1 (*ACAT-1*) gene expression of macrophages isolated from streptozotocin-induced type 1 diabetes (T1D) mice and T1D patients as well as advanced glycation end product (AGE)-exposed mouse peritoneal macrophages and THP-1 cells. Foam cell formation, *CD36* and *ACAT-1* gene expression of macrophages derived from T1D mice or patients increased compared with those from non-diabetic controls, all of which were inhibited by 10 nmol/L teneligliptin. AGEs mimicked the effects of T1D; teneligliptin attenuated all the deleterious effects of AGEs in mouse macrophages and THP-1 cells. Our present findings suggest that teneligliptin may inhibit foam cell formation of macrophages in T1D *via* suppression of *CD36* and *ACAT-1* gene expression partly by attenuating the harmful effects of AGEs.

## 1. Introduction

Diabetes is associated with an increased risk of atherosclerotic cardiovascular disease, and half of diabetic patients die from this devastating disorder [1]. Various biochemical pathways are activated under diabetic conditions, thereby being involved in the development and progression of atherosclerosis [2,3]. Among them, advanced glycation end products (AGEs), senescent macromolecule derivatives formed at an accelerated rate under hyperglycemic and oxidative stress conditions, play a crucial role in atherosclerotic cardiovascular disease of patients with type 1 diabetes (T1D) and type 2 diabetes (T2D) [4,5,6]. Indeed, AGEs are localized in macrophage-derived foam cells within the atherosclerotic lesions and associated with endothelial dysfunction and arterial stiffness, being a prognostic marker of future cardiovascular events in T1D and T2D patients [7,8,9,10,11,12].

Accumulation of cholesterol esters and foam cell formation of macrophages are one of the early characteristic features of atherosclerosis [2,3], which are partly dependent on uptake of oxidized low-density lipoprotein (ox-LDL) via scavenger receptor CD36 [13] and esterification of free cholesterol to cholesteryl ester by acyl-coenzyme A: cholesterol acyltransferase-1 (ACAT-1) [13]. Furthermore, the foam cell formation of macrophages is enhanced under diabetic states [14,15,16], which could contribute to the increased risk of macrovascular complications in diabetes [17].

Dipeptidyl peptidase-4 (DPP-4) inhibitors have been known to improve hyperglycemia in T2D patients by stimulating the incretin effects via suppressing the degradation of incretins, such as glucagon-like peptide-1 (GLP-1) and glucose-dependent insulinotropic polypeptide (GIP) [18]. DPP-4 inhibitors are now one of the widely used drugs for the treatment of T2D patients due to its low risk of weight gain and hypoglycemia [18]. Recently, DPP-4 inhibitors have been shown to play a protective role against atherosclerosis in diabetic animal models [17,19] and T2D patients [20,21,22]. Indeed, we have found that foam cell formation and atherosclerotic lesions of diabetic apolipoprotein E-deficient mice were significantly increased compared with non-diabetic counterparts, which were prevented by vildagliptin [17]. Furthermore, an inhibitor of DPP-4, teneligliptin has been shown to significantly attenuate ox-LDL uptake and foam cell formation of macrophages isolated from obese T2D mice and patients partly by suppressing *CD36* and *ACAT-1* expression [14]. However, since T2D is associated with dyslipidemia, hypertension, and insulin resistance, part of which are ameliorated by DPP-4 inhibitors [18], it remains unclear whether DPP-4 inhibitors could have anti-atherosclerotic properties directly by blocking the harmful effects of hyperglycemia or indirectly by ameliorating these comorbidities. On the other hand, we have previously found that DPP-4 inhibitors suppress atherosclerotic vascular injury in diabetic animals by inhibiting the deleterious effects of AGEs [18,23,24,25,26]. These findings led us to speculate that DPP-4 inhibitors could attenuate atherosclerosis partly by suppressing the harmful effects of AGEs on macrophages. To address the issue, we examined here whether teneligliptin could inhibit ox-LDL uptake, foam cell formation, *CD36* and *ACAT-1* gene expression of macrophages isolated from streptozotocin-induced T1D mice and T1D patients as well as AGE-exposed mouse peritoneal macrophages and THP-1 cells.

## 2. Results

### 2.1. Characteristics and Laboratory Data of Mice and Humans

Laboratory data of *wild-type* mice and streptozotocin (STZ)-induced T1D mice are presented in Table 1. Compared with the *wild-type* mice, T1D mice exhibited severe hyperglycemia, low body weight, and few insulin levels with marked elevation of glycated hemoglobin (HbA1c). The area under the curve of blood glucose during oral glucose tolerance test (OGTT) was significantly higher in T1D mice than *wild-type* mice (Table 1).

Table 2 summarizes the clinical characteristics of five T1D patients and six volunteers. Blood glucose and HbA1c values were significantly higher in T1D patients than controls. Fasting C-peptide and stimulated C-peptide levels were dramatically decreased in T1D patients. The number of patients with simple retinopathy, stage 2–3 diabetic nephropathy, and peripheral artery disease are 1, 2, and 1, respectively. Two T1D patients received statins for dyslipidemia, while 3 anti-hypertensive agents for hypertension.

### 2.2. Teneligliptin Suppressed Foam Cell Formation of Macrophages Isolated from T1D Mice and T1D Patients

We first evaluated the effects of teneligliptin on ox-LDL uptake and foam cell formation of macrophages isolated from T1D mice and *wild-type* mice. Immunofluorescent staining showed that 1,1'-dioctadecyl-3,3,3',3'-tetramethylindocarbocyanine perchlorate-oxidized-low-density lipoprotein (Dil-ox-LDL)-positive cells were co-stained with F4/80, a marker of macrophages (Figure 1A–I), confirming the uptake of ox-LDL into mouse peritoneal macrophages. As shown in Figure 1J, Dil-ox-LDL uptake into macrophages was significantly increased in T1D mice compared with *wild-type* mice, which was completely prevented by the treatment with 10 nmol/L teneligliptin. When the foam cell formation of macrophages was evaluated by cholesterol esterification assay using ox-LDL with [^3^H]oleate, foam cell formations of macrophages isolated from T1D mice and T1D patients were significantly increased in comparison with controls, both of which were attenuated by 10 nmol/L teneligliptin (Figure 1K,L). Furthermore, 10 nmol/L teneligliptin significantly inhibited the up-regulation of *CD36* and *ACAT-1* gene expression in macrophages derived from T1D mice and T1D patients (Figure 1M–P).

### 2.3. Teneligliptin Inhibited ox-LDL Uptake in AGE-exposed Mouse Macrophages and Human THP-1 Macrophages

We further evaluated the effects of teneligliptin on ox-LDL uptake in AGE-exposed mouse peritoneal macrophages and human THP-1 cells. As shown in Figure 2A–H, AGE-bovine serum albumin (AGE-BSA) significantly increased ox-LDL uptake into mouse peritoneal macrophages and human THP-1 cells compared to non-glycated control BSA, which were completely prevented by 10 nmol/L teneligliptin. In addition, 10 nmol/L teneligliptin completely suppressed the up-regulation of *CD36* and *ACAT-1* gene expression in AGE-exposed mouse peritoneal macrophages and human THP-1 cells (Figure 2I–L).

## 3. Discussion

DPP-4 inhibitors have been shown to play atheroprotective properties in both animal models and patients with T2D [17,19,20,21,22]. Indeed, although large clinical trials with DPP-4 inhibitors did not reduce the risk of major cardiovascular events [27,28,29,30], a couple of clinical papers reported that DPP-4 inhibitors showed favorable effects on carotid intima-media thickness in T2D patients, a surrogate marker of atherosclerosis, and sitagliptin use was independently associated with the lower incidence rates of atherosclerotic cardiovascular disease in subjects with T2D [20,21,22]. DPP-4 inhibitors not only ameliorate hyperglycemia, but also improve dyslipidemia with modest blood pressure lowering effects [31,32,33]. Furthermore, there is an accumulating body of evidence to show that GLP-1 and GIP have pleiotropic effects on diabetic vessels; GLP-1 and GIP could attenuate oxidative stress generation, inflammatory reactions, and foam cell formation of macrophages within the atherosclerotic plaques [34,35,36,37], thus playing a protective role against atherosclerosis. Therefore, it remains unclear whether DPP-4 inhibitors slow the process of atherosclerosis directly *via* the improvement of hyperglycemia and related AGE formation, indirectly *via* the amelioration of comorbid risk factors, or through the pleiotropic effects of incretins. To address whether the effects of DPP-4 inhibitor are partly independent of these comorbidities associated with T2D and pleiotropic actions of incretins, we first examined the effects of a DPP-4 inhibitor, teneligliptin on ox-LDL uptake and foam cell formation of macrophages isolated from STZ-induced T1D mice and T1D patients because (1) blood pressure levels and lipid parameters in T1D mice and patients were comparable with those in controls, (2) body weight, triglycerides, and homeostasis model assessment of insulin resistance (HOMA-IR), a marker of insulin resistance were higher in the previously published T2D mice models or patients than controls, (3) there were significant differences of triglycerides and HOMA-IR or fasting C-peptide between T1D and T2D, and (4) DPP-4 inhibitors did not modulate the effects of incretins on macrophages in vitro. In other words, we used here macrophages from T1D model mice and patients to rule out the effects of these comorbidities.

In this study, we found for the first time that teneligliptin at 10 nmol/L significantly blocked ox-LDL uptake and foam cell formation of macrophages isolated from STZ-induced T1D mice or T1D patients. Furthermore, teneligliptin significantly inhibited the up-regulation of *CD36* and *ACAT-1* mRNA levels in macrophages isolated from T1D mice and T1D patients. CD36 is a major scavenger receptor mediating uptake of ox-LDL into macrophages, whereas ACAT-1 is a rate-limiting enzyme for the esterification of free cholesterol, which could contribute to macrophage foam cell formation [13]. Therefore, our present observations suggest that teneligliptin could inhibit ox-LDL uptake and foam cell formation of macrophages in T1D partly via suppression of *CD36* and *ACAT-1* expression. The present findings have extended the previous observations showing that DPP-4 inhibitors directly suppressed foam cell formation of macrophages in vitro partly via inhibition of CD36 [14,38]. The peak plasma concentration of teneligliptin after oral administration of clinical dose of 20 mg is about 280 nmol/L, and approximately 20% of the amount of teneligliptin in the blood is a protein-unbound, free form [39]. Therefore, the concentration of teneligliptin (10 nmol/L) with beneficial effects on macrophages observed here may be comparable to the therapeutic level (less than 50 nmon/L) which is achieved in the treatment for patients with T2D. Since ox-LDL uptake and foam cell formation are key components of atherosclerosis [2,3], our present study suggests that teneligliptin may inhibit the progression of atherosclerosis through its direct pleiotropic effects on macrophages.

We have already reported that teneligliptin significantly reduced ox-LDL uptake, foam cell formation, *CD36/ACAT-1* mRNA levels of macrophages derived from T2D patients and *db/db* mice, an animal model of T2D [14]. However, it remains unclear how teneligliptin directly inhibits ox-LDL uptake and foam cell formation of macrophages. In other words, the mechanisms of action of the drug are not fully elucidated. Therefore, we examined the effects of teneligliptin on ox-LDL uptake, *CD36* and *ACAT-1* gene expression in AGE-exposed mouse peritoneal macrophages and THP-1 cells, a human macrophage cell line because (1) we have previously shown that DPP-4 inhibitors could block the harmful effects of AGE in cultured endothelial cells and renal proximal tubular cells [24,40] and (2) AGE play a central role in atherosclerosis in diabetes [18]. In this study, we found that AGE mimicked the effects of T1D; AGE significantly increased ox-LDL uptake, *CD36* and *ACAT-1* gene expression in mouse macrophages and THP-1 cells, all of which were prevented by the treatment with 10 nmol/L teneligliptin. Therefore, teneligliptin could suppress the ox-LDL uptake and foam cell formation of macrophages isolated from T1D mice and patients partly *via* suppression of deleterious effects of AGE on macrophages. AGE and macrophages were co-existed within the atherosclerotic plaques, which were associated with the severity of atherosclerosis [7,8,41]. In addition, circulating and tissue accumulation levels of AGE were associated with the increased risk of cardiovascular disease in both diabetic and non-diabetic patients [5,6,10,11,42,43,44,45,46]. These findings further support the clinical relevance of anti-atherosclerotic effects of teneligliptin in diabetes.

The present study has some potential limitations. First, we examined the atheroprotective role of teneligliptin only focusing on macrophage foam cell formation in vitro. Second, it is probable that any of clinical characteristics of T1D patients, including disease duration, age and drug medications, such as statins and anti-hypertensive agents could have impacted the present findings. Indeed, in this study, age and sex were not completely matched between T1D cases and controls. Since accumulation of AGE has been known to progress in a normal ageing process and under diabetic condition, especially diabetic patients with a long disease history [45], the difference of mean ages of the T1D cases and controls may influence the present results. In addition, there is a sex disparity in cardiovascular event and mortality rates associated with diabetes [47], and therefore the difference of number of male/female in cases/controls may also affect the ox-LDL uptake and foam cell formation of macrophages. However, we could not sub-analyze the data because of small number of patients in this study. Third, in order to avoid the effects of streptozotocin and narrow it down the specific effect of hyperglycemia on macrophages, we investigated the effects of AGE, a marker of cumulative hyperglycemic exposure on ox-LDL uptake, *CD36* and *ACAT-1* gene expression of macrophages derived from non-diabetic mice. Fourth it would be relevant to see the impact of teneligliptin on ox-LDL, foam cell formation and *CD36/ACAT-1* mRNA levels under normal glycemic conditions. Since teneligliptin is an inhibitor of DPP-4, which is approved for the treatment of diabetes, we examined here the effects of teneligliptin on macrophages under diabetic conditions. Fifth, we performed the present experiments at only 1 dose of teneligliptin (10 nmol/L). It would be interesting to examine the dose-dependent effects of teneligliptin on macrophages. Sixth, it would be valuable to demonstrate CD36 and ACAT-1 protein levels by western blotting. However, CD36 and ACAT-1 protein expression levels are functionally correlated with ox-LDL uptake and foam cell formation of macrophages, respectively. The observation suggests that these gene expression levels would be correlated with protein levels. Seventh, it would be interesting to explore other genes, such as interleukin-6 with changing expression levels by teneligliptin through RNA-sequencing. Eighth, although Xu et al. reported that AGE at supraphysiological concentrations (300–600 µg/mL AGE-BSA) increased lipid accumulation in macrophages partly by regulating CD36, scavenger receptor A2, hydroxymethylglutaryl-CoA reductase, ACAT-1, and ATP-binding cassette transporter G1 [48], we provided here a new line of following evidence; concentration of AGE (100 µg/mL AGE-BSA), which is comparable with that of in vivo-diabetic situation [49,50], actually stimulated ox-LDL uptake of macrophages via *CD36* and *ACAT-1* expression, which is a molecular target for atheroprotective properties of teneligliptin. Ninth, intraperitoneal administration of AGE-BSA has been reported to impair glucose tolerance in mice in association with decrease in acute insulin secretion [51], AGE may augment foam cell formation of macrophages by further deteriorating hyperglycemia. Specifically, the salient findings of our present study are that teneligliptin at a therapeutic level inhibited foam cell formation of macrophages by suppressing the harmful effects of AGE, whose concentration is also comparable with that of diabetic conditions. In any case, further clinical studies should be needed to clarify whether DPP-4 inhibitors could inhibit foam cell formation of macrophages and resultantly reduce the risk of cardiovascular events in patients with diabetes.

## 4. Materials and Methods

### 4.1. Animal Experiments

STZ was purchased from Sigma–Aldrich (St. Louis, MO, USA) and a DPP-4 inhibitor teneligliptin was generously gifted by Tanabe Mitsubishi Pharma (Tokyo, Japan). Animal experiments were conducted under strict accordance with the recommendations in the Guide for the Care and Use of Laboratory Animals [52]. The study design was approved by the Animal Care Committee of Showa University (permission number: 07005). All surgeries and sacrifices were performed under general anesthesia using isoflurane and with efforts to minimize the suffering.

A total of 12 male *C57BL/6J (wild-type)* mice at 7 weeks old were purchased from Sankyo Labo Service (Tokyo, Japan), kept on a standard rodent chow (Labo MR Stock, NOSAN, Yokohama, Kanagawa, Japan) with free access to water, and housed within a specific pathogen-free facility in the Division of Animal Experimentation of Showa University School of Medicine. The rooms were controlled under a 12-h dark/light cycle, 21 °C temperature, and 40–60% humidity. At 8 weeks old, the mice received intraperitoneal injections of saline or STZ (50 mg/kg/day) for 5 consecutive days to create a T1D model mouse, as previously described [17]. All mice showed FBG levels higher than 200 mg/dL, and were used in the present experiments as T1D model mice. All mice did not show any clinical signs, including severe weight loss by more than 20% from the baseline or obvious weakness. At 13 weeks old, blood samples were collected, and peritoneal macrophages were extracted from mice after intraperitoneal injection of thioglycolate broth as previously described [14,17,31,36,37].

### 4.2. Measurements of Laboratory Parameters in Mice

Blood samples collected after a 12-h fast were used for the analysis of biochemical analysis. Systolic and diastolic blood pressures (SBP and DBP) were measured, and FBG, HbA1c, total cholesterol (Total-C), high-density lipoprotein cholesterol (HDL-C), triglycerides, and insulin levels were measured, and OGTT was performed as previously described [14,17].

### 4.3. Experiments of Human Macrophages

The study protocol was approved by the Ethics Committee of Showa University School of Medicine (Tokyo, Japan; approval number: 2799). Written informed consent was obtained from all T1D patients and healthy volunteers. The study was designed in compliance with the Declaration of Helsinki.

Five patients with uncontrolled T1D despite multiple daily insulin injections over ≥12 weeks and six controls were enrolled in the present study. Blood samples were collected, and human monocyte-derived macrophages were isolated using anti-CD14 antibody-conjugated magnetic microbeads (Miltenyi Biotec, Auburn, CA, USA) as previously described [14].

### 4.4. Measurements of Clinical Parameters in Humans

Body mass index (BMI), SBP and DBP were measured and FBG, HbA1c (NGSP), LDL-C, HDL-C, triglycerides, fasting C-peptide and the 6-min value of C-peptide after glucagon (1 mg) stimulation test (stimulated C-peptide) were measured by standard methods as described previously [53].

### 4.5. Preparation of AGE-BSA

AGE-BSA was prepared as previously described [54]. In brief, BSA (25 mg/mL) was incubated under sterile conditions with 0.1 M glyceraldehyde in 0.2 M NaPO4 buffer (pH 7.4) for 7 days. Control non-glycated BSA was incubated in the same conditions except for the absence of reducing sugars.

### 4.6. Differentiation of THP-1 Macrophages

A human monocytic cell line, THP-1 cells were maintained in RPMI 1640 medium with 10% fetal bovine serum. The cells were seeded onto 3.5-cm dishes (1.0 × 10^6^ cells/dish) and incubated with phorbol 12-myristate 13-acetate (40 ng/mL; Sigma–Aldrich, St. Louis, MO, USA) for 24 h to differentiate macrophages [55].

### 4.7. Immunofluorescent Staining of Mouse Macrophages and THP-1 Macrophages

Peritoneal macrophages isolated from *wild-type* mice and T1D model mice, or THP-1 macrophages were incubated with or without 100 µg/mL AGE-BSA or 100 µg/mL non-glycated BSA for 24 h, and then treated with 10 μg/mL Dil-ox-LDL in the presence or absence of 10 nmol/L teneligliptin for 18 h. After washing, they were mounted in Vectashield mounting medium (H-1500, Vector Laboratories, Burlingame, CA, USA) and were imaged with BZ-X710 microscope/software (Keyence, Osaka, Japan) as previously described [14].

### 4.8. Cholesterol Esterification Assay in Macrophages Isolated from Mice and Humans

Cholesterol esterification assay was conducted as previously described [14,17,31,36,37]. In brief, peritoneal macrophages isolated from *wild-type* mice and T1D model mice or human monocyte-derived macrophages from T1D patients and controls were incubated with 10 μg/mL ox-LDL and 0.1 mmol/L [^3^H]oleate for 18 h in the presence or absence of 10 nmol/L teneligliptin [14]. Cellular lipids were extracted and the radioactivity of cholesterol [^3^H]oleate was determined by a thin-layer chromatography.

### 4.9. Gene Expression Levels in Macrophages Isolated from T1D Mice and T1D Patients and in AGE-Exposed Cells

Peritoneal macrophages isolated from *wild-type* and T1D model mice, monocyte-derived macrophages from T1D patients and controls, or THP-1 macrophages were treated with or without 100 µg/mL AGE-BSA or 100 µg/mL non-glycated BSA for 24 h, and then incubated with or without 10 nmol/L teneligliptin for 18 h [14]. Total RNA was isolated with QIAGEN reagents (Hilden, Germany), and gene expression was analyzed by real-time reverse-transcription polymerase chain reactions using the TaqMan gene expression assay and a sequence detection system (ABI PRISM 7900, Life Technologies, Thermo Fischer Scientific, Pleasanton, CA, USA) as previously described [14]. The pre-designed TaqMan probe sets used in mice were as follows: *Cd36*, Mm01135198_ml; *Acat-1*, Mm00507463_ml; glyceraldehyde-3-phosphate dehydrogenase (*Gapdh*), Mm03302249_g1. Human probes were as follows: *Cd36*, Hs00169627_ml; *Acat-1*, Hs01009746_ml; *Gapdh*, Hs99999905_ml.

### 4.10. Statistical Analysis

Values were expressed as mean ± standard deviation. Statistical analyses were performed by unpaired *t*-test to compare two groups and analysis of variance (ANOVA) to compare multiple groups followed by appropriate multiple comparison tests. Categorical variables were compared by chi-square test. All analyses were performed by PRISM 7.0 software (GraphPad, San Diego, CA, USA). The significance level was defined as *p* < 0.05.

## 5. Conclusions

The present study suggests that teneligliptin could inhibit foam cell formation of macrophages in T1D *via* suppression of CD36 and ACAT-1 gene expression partly by attenuating the harmful effects of AGEs.

## Figures and Tables

**Figure 1 ijms-21-04811-f001:**
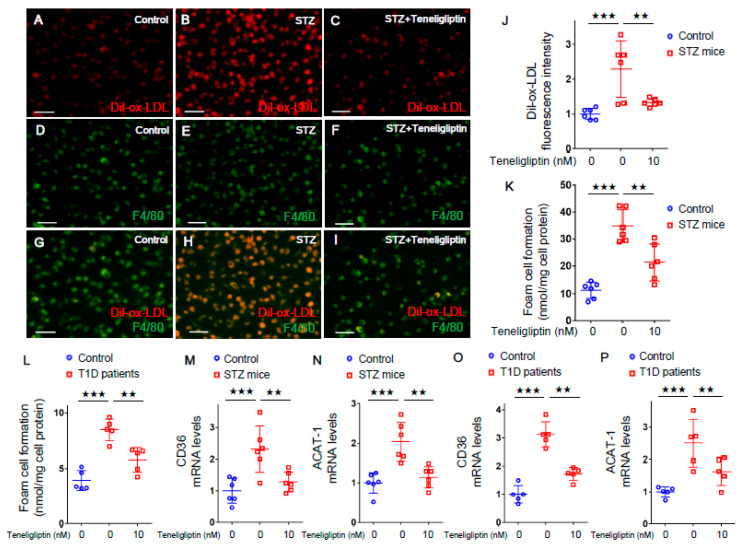
Effects of teneligliptin on oxidized low-density lipoprotein (ox-LDL) uptake, foam cell formation, and *CD36* and *ACAT-1* gene expression in macrophages extracted from type 1 diabetes (T1D) model mice and T1D patients. (**A–I**) Representative immunofluorescent staining images in the peritoneal macrophages isolated from *wild-type* mice and T1D model mice. Dil-ox-LDL staining cells were in red, and F4/80 expressing cells were in green. Scale bars represent 50 µm. (**J**) Fluorescence intensity of Dil-ox-LDL per area. (**K**) and (**L**) Foam cell formation was evaluated by the radioactivity of cholesterol [^3^H] oleate. (**M–P**) Gene expression levels of *CD36* (**M**,**O**) and *ACAT1* (**N**,**P**) in peritoneal macrophages isolated from mice and in monocyte-derived macrophages from humans. (**J**,**M–P**) are normalized to the control levels.(**A–K**,**M**,**N**); number = 6 for each group. L, O, and P, number = 6 for control group. number = 5 for each T1D group. *** *p* < 0.005, ** *p* < 0.01.

**Figure 2 ijms-21-04811-f002:**
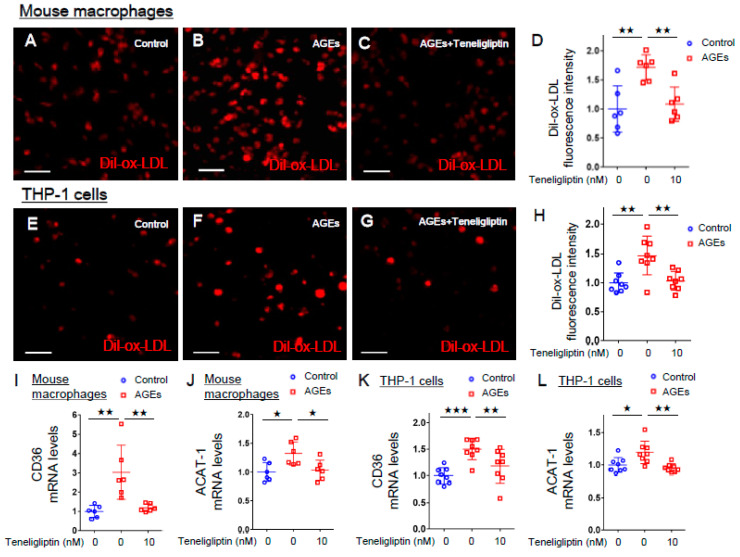
Effects of teneligliptin on ox-LDL uptake, *CD36* and *ACAT-1* gene expression in AGE-exposed mouse macrophages and THP-1 cells. (**A**–**C**,**E**–**G**) Representative immunofluorescent staining images in mouse peritoneal macrophages (**A**–**C**) and THP-1 cells (**E**–**G**). Dil-ox-LDL staining cells were in red. Scale bars represent 50 µm for A-C and 100 µm for E-G. (**D**,**H**) Fluorescence intensity of Dil-ox-LDL per area. (**I–L**) Gene expression levels of *CD36* (**I**,**K**) and *ACAT-1* (**J**,**L**) in mouse macrophages and THP-1 cells. (**A–C,D,I,J**) number = 6 for each group. (**E–G,H,K,L**) number = 8 for each group. (D,H-L) are normalized to the control levels. *** *p* < 0.005, ** *p* < 0.01, * *p* < 0.05.

**Table 1 ijms-21-04811-t001:** Laboratory characteristics of *wild-type* mice and streptozotocin-induced type 1 diabetes mice at 13 weeks old.

	*Wild-Type* Mice	T1D Model Mice	*p*-Value
Number	6	6	−
Final body weight (g)	24.3 ± 1.6	20.3 ± 3.2	*P* < 0.01 ^★^
Food Intake (g/day)	4.5 ± 0.4	4.8 ± 1.0	0.459
SBP (mmHg)	100 ± 16	102 ± 15	0.840
DBP (mmHg)	61 ± 6	65 ± 7	0.302
Total-C (mg/dL)	74 ± 9	86 ± 22	0.592
HDL-C (mg/dL)	40 ± 17	32 ± 15	0.378
Triglycerides (mg/dL)	60 ± 9	82 ± 39	0.205
FBG (mg/dL)	90 ± 8	164 ± 56	*p* < 0.01 ^★^
Insulin (ng/mL)	0.27 ± 0.08	0.05 ± 0.02	*p* < 0.001 ^★^
HbA1c (%)	4.3 ± 0.2	7.9 ± 0.5	*p* < 0.001 ^★^
OGTT-AUC of glucose(mg/dL x hour)	608 ± 36	1150 ± 365	*p* < 0.005 ^★^

T1D, type 1 diabetes; SBP, systolic blood pressure; DBP, diastolic blood pressure; Total-C, Total cholesterol; HDL-C, high-density lipoprotein cholesterol; FBG, fasting blood glucose; HbA1c, glycated hemoglobin; OGTT, oral glucose tolerance test; AUC, Area under the curve. Results are presented as mean values ± SD and analyzed with unpaired t-test. ^★^
*p* < 0.05 *vs*. *C57BL6/J* mice.

**Table 2 ijms-21-04811-t002:** Clinical parameters of type 1 diabetes patients and controls.

	Controls	T1D Patients	*p*-Value
Number (male/female)	6 (5/1)	5 (3/2)	0.251
Age (years)	42 ± 10	58 ± 30	0.25
Duration of diabetes (years)	−	11 ± 10	−
Body Weight (kg)	67 ± 8	71 ± 20	0.725
BMI (kg/m^2^)	23.1 ± 1.3	21.3 ± 3.9	0.318
SBP (mmHg)	114 ± 5	114 ± 10	0.912
DBP (mmHg)	71 ± 7	68 ± 13	0.662
Total-C (mg/dL)	187 ± 7	167 ± 24	0.087
LDL-C (mg/dL)	110 ± 38	92 ± 28	0.11
HDL-C (mg/dL)	52 ± 14	57 ± 20	0.657
Triglycerides (mg/dL)	110 ± 38	90 ± 48	0.465
FBG (mg/dL)	93 ± 3	233 ± 78	p < 0.005 ^★^
HbA1c (%)	5.2 ± 0.4	7.7 ± 0.4	p < 0.001 ^★^
Fasting C-peptide (ng/mL)	N.A.	0.16 ± 0.09	−
Stimulated C-peptide (ng/mL)	N.A.	0.28 ± 0.25	−
Retinopathy (NDR/SDR/PPDR/PDR)	N.A.	(4/1/0/0)	−
Nephropathy (1/2/3/4/5)	N.A.	(3/1/1/0/0)	−
PAD/none	N.A.	(1/4)	−
Total daily insulin dose (Unit)	−	48 ± 36	−
Lipid-lowering drugs (statins/none)	(1/5)	(2/3)	0.251
Anti-hypertensive drugs(ARBs/ARBs+CCBs/CCBs/none)	(0/0/0/6)	(1/1/1/2)	0.176

T1D, type 1 diabetes; BMI, body mass index; SBP, systolic blood pressure; DBP, diastolic blood pressure; Total-C, total cholesterol; LDL-C, low-density lipoprotein cholesterol; HDL-C, high-density lipoprotein cholesterol; FBG, fasting blood glucose; HbA1c, glycated hemoglobin; NDR, no diabetic retinopathy; SDR, simple diabetic retinopathy; PPDR, pre-proliferative diabetic retinopathy; PDR, proliferative diabetic retinopathy; N.A., not available; −, none; PAD, peripheral artery disease; ARB, angiotensin II receptor blockers; CCBs, calcium channel blockers. Results are presented as mean values ± SD and analyzed with unpaired t-test. Categorical variables were compared by chi-square test. ^★^
*p* < 0.05 *vs*. controls.

## Data Availability

All data used to support the findings of this study are available from the corresponding author upon reasonable request.

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
