# Peer review of "A Dipeptidyl Peptidase-4 Inhibitor Inhibits Foam Cell Formation of Macrophages in Type 1 Diabetes via Suppression of CD36 and ACAT-1 Expression"

_ijms, 2020, doi:10.3390/ijms21134811_

Round 1

Reviewer 1 Report

In the submitted manuscript, Terasaki et al describe the use of the DPP-4 inhibitor, teneligliptin, to inhibit foam cell formation of macrophages derived from (i) the streptozotocin-induced mouse model of T1D, (ii) human patients living with T1D, and (iii) the THP-1 human monocytic cell line.

The study's aim was to investigate whether the mechanism of foam cell inhibition was linked to reduced CD36 and ACAT-1 expression in macrophages via the effect of teneligliptin on the AGE–receptor axis; thus providing evidence for a direct role for DPP-4 inhibitors in attenuating the consequences of hyperglycaemia rather than improving co-morbidities of diabetes (e.g. dyslipidemia, hypertension, and insulin resistance).

Strengths: The investigation combined evidence from animal models, human patients and human-derived cell lines reflecting actual, induced and simulated T1D. The ex vivo concentration of teneligliptin applied herein reflects a plausible therapeutic dose used in the treatment of T2D.

Specific points to address:

  • The human subjects are described as 5 T1D patients and 6 age-gender matched controls. How can the controls be matched for age and gender when there are not even numbers in each group? Please provide the gender breakdown (e.g. male = n). The duration of T1D must vary significantly with a mean of 11 years and SD of 10 years. Could you please comment in the discussion as to whether any of the clinical characteristics of the T1D patients (including disease duration) could have impacted ox-LDL uptake, foam cell formation, or expression of CD36 or ACAT-1?
  • It is not clear how the experiments summarised in Figure 1 were performed. The manuscript describes that macrophages isolated from STZ-treated mice and human T1D patients showed increased uptake of ox-LDL and foam cell formation (as measured by cholesterol esterification), as well as increased CD36 and ACAT-1 mRNA levels, compared with macrophages of untreated mice and healthy human subjects. For both the mice and humans, treatment with 10 nmol/L teneligliptin rescued the phenotype. There is no description in the method section, however, outlining the incubation conditions for these primary macrophages. Sections 4.7 and 4.9 of the method section apply to data presented in Figure 2. Section 4.8 covers Figures 1(K) and 1(L), only. This should be addressed.
  • It is also not clear why the authors have only presented data on teneligliptin treatment of their “experimental” groups (e.g. T1D patients, STZ-mice, AGE-exposed cell lines). It would be relevant to see the impact of teneligliptin on ox-LDL, foam cell formation and CD36/ACAT-1 under normo-glycaemic conditions. If the authors don’t think this is relevant, please explain why. Likewise, only 1 does of teneligliptin was included. Have the authors seen a dose-dependent effect?
  • The micrographs presented in Figure 1(A-I) and 2(A-G) are extremely difficult to interpret. Potentially the resolution of the provided image is too low, however, it would also appear the magnification is low. Are any images at higher magnification available?
  • It appears results for Figures 1(J,M-P) and Figures 2(D-L) have been normalised to the control levels. If this is the case, it should be stated. Moreover, the columns included in the bar charts are not relevant in interpreting these data and should be removed. The range indicated by the error bars should be stipulated (e.g. SD, IQR, other?).
  • The authors sensibly point out that only CD36 and ACAT-1 mRNA levels were measured. It would be highly valuable to demonstrate this at the protein level by Western blotting. Antibodies targeting both these proteins should be readily available.
  • It would also be interesting to explore other genes with changing expression levels following teneligliptin exposure, for example, through RNA-seq.
  • The use of the term “Anyway,” in line 196 is very dismissive. Consider substituting for “In any case,”
  • The formatting of some references is not consistent. The titles of ref 41 and 47 are not underlined. The authors but not the title of ref 49 are underlined.

Author Response

Response to the Reviewers’ comments

Reviewer #1

  1. The human subjects are described as 5 T1D patients and 6 age-gender matched controls. How can the controls be matched for age and gender when there are not even numbers in each group? Please provide the gender breakdown (e.g. male = n).

Thank you for your comments. As you pointed out, controls and T1D patients were not even numbers. However, there were no significant differences of age and gender between the two groups by chi-square test. We provided the number of male and female in each group in the revised Table 2.

  1. The duration of T1D must vary significantly with a mean of 11 years and SD of 10 years. Could you please comment in the discussion as to whether any of the clinical characteristics of the T1D patients (including disease duration) could have impacted ox-LDL uptake, foam cell formation, or expression of CD36 or ACAT-1?

Thank you for your comments. As you pointed out, it is probable that any of the clinical characteristics of the T1D patients, including disease duration, could have impacted the present findings. However, we could not sub-analyze the data because of small number of patients in the present study. We added these statements in the last paragraph of Discussion section of revised manuscript.

  1. It is not clear how the experiments summarised in Figure 1 were performed. The manuscript describes that macrophages isolated from STZ-treated mice and human T1D patients showed increased uptake of ox-LDL and foam cell formation (as measured by cholesterol esterification), as well as increased CD36 and ACAT-1 mRNA levels, compared with macrophages of untreated mice and healthy human subjects. For both the mice and humans, treatment with 10 nmol/L teneligliptin rescued the phenotype. There is no description in the method section, however, outlining the incubation conditions for these primary macrophages. Sections 4.7 and 4.9 of the method section apply to data presented in Figure 2. Section 4.8 covers Figures 1(K) and 1(L), only. This should be addressed.

Thank you for your comments. I am wandering if you have a little confusion. We have already described the incubation conditions for Figure 1 as follows.

・Peritoneal macrophages were extracted from control mice and T1D mode mice after intraperitoneal injection of thioglycollate broth (Sections 4.1).

・Human monocyte-derived macrophages from controls and T1D patients were isolated using CD14 antibody-conjugated magnetic microbeads (Sections 4.3).

・Section 4.7 covers Figure 1(J), while section 4.9 covers Figures 1(M), 1(N), 1(O) and 1(P).

  1. It is also not clear why the authors have only presented data on teneligliptin treatment of their “experimental” groups (e.g. T1D patients, STZ-mice, AGE-exposed cell lines). It would be relevant to see the impact of teneligliptin on ox-LDL, foam cell formation and CD36/ACAT-1 under normo-glycaemic conditions. If the authors don’t think this is relevant, please explain why.

Thank you for your comments. We totally agreed with your opinion. It would be relevant to see the impact of teneligliptin on ox-LDL, foam cell formation and CD36/ACAT-1 mRNA levels under normal glycemic conditions. However, since teneligliptin is an inhibitor of DPP-4, which is approved for the treatment of diabetes, we examined here the effects of teneligliptin on macrophages under diabetic conditions. We added these statements in the last paragraph of Discussion section of revised manuscript.

  1. Likewise, only 1 does of teneligliptin was included. Have the authors seen a dose-dependent effect?

Thank you for your comments. Unfortunately, we did not know a dose-dependent effect of teneligliptin. It would be interesting to examine the dose-dependent effect of teneligliptin on macrophages. We added these statements in the last paragraph of Discussion section of revised manuscript.

  1. The micrographs presented in Figure 1(A-I) and 2(A-G) are extremely difficult to interpret. Potentially the resolution of the provided image is too low, however, it would also appear the magnification is low. Are any images at higher magnification available?

Thank you for your comments. According to your request, we provided new images with high resolution and magnification in the revised Figures 1(A-I) and 2 (A-G).

  1. It appears results for Figures 1(J,M-P) and Figures 2(D-L) have been normalised to the control levels. If this is the case, it should be stated. Moreover, the columns included in the bar charts are not relevant in interpreting these data and should be removed. The range indicated by the error bars should be stipulated (e.g. SD, IQR, other?).

 Thank you for your comments. As you pointed out, Figures 1(J,M-P) and Figures 2(D-L) have been normalized to the control levels. Therefore, it was stated in the legends of Figure 1 and 2. Furthermore, according to your comments, bar charts were removed and error bars (SD) were stipulated in the revised Figures 1 and 2.

  1. The authors sensibly point out that only CD36 and ACAT-1 mRNA levels were measured. It would be highly valuable to demonstrate this at the protein level by Western blotting. Antibodies targeting both these proteins should be readily available.

Thank you for your comments. We totally agreed with your opinion. It would be highly valuable to evaluate CD36 and ACAT1 protein levels by Western blotting. However, CD36 and ACAT-1 protein expression levels are functionally correlated with ox-LDL uptake and foam cell formation of macrophages, respectively. The findings suggest that these gene expression levels would be correlated with protein levels. We added these statements in the last paragraph of Discussion section of revised manuscript.

  1. It would also be interesting to explore other genes with changing expression levels following teneligliptin exposure, for example, through RNA-seq.

Thank you for your comment. As you pointed out, it would be interesting to explore other genes with changing expression levels by teneligliptin through RNA-seq. We added these statements in the last paragraph of Discussion section of revised manuscript.

  1. The use of the term “Anyway,” in line 196 is very dismissive. Consider substituting for “In any case,”

Thank you for your comment. According to your request, we changed the word “Anyway” to “In any case” in the revised manuscript.

  1. The formatting of some references is not consistent. The titles of ref 41 and 47 are not underlined. The authors but not the title of ref 49 are underlined.

Thank you for your comments. According to your request, we corrected the style of references in the revised manuscript.

Thank you again for your kindness.

Sincerely yours,

                           Michishige Terasaki, M.D., Ph.D.

Division of Diabetes, Metabolism, and Endocrinology

Department of Medicine

Showa University School of Medicine

Tokyo 142-8666, Japan

Tel: +81-3-3784-8000

Fax: +81-3-3784-8948

E-mail: ttmichishige@yahoo.co.jp

Reviewer 2 Report

The manuscript “A dipeptidyl peptidase-4 inhibitor inhibits foam cell formation of macrophages in type 1 diabetes via suppression of CD36 and ACAT-1 expression” presents the data on the effect of the drug teneligliptin on the expression of effector genes CD36 and ACAT-1 in type 1 diabetes induced mice and type 1 diabetes patients ex vivo.

The literature around the issue has evolved and part of the evidence has come from the research by the authors of this manuscript. ACAT-1 was considered as one of the key elements in atherosclerosis where it was shown for example upregulation of ACAT-1 in dexamethasone long term users possibly by  enhancing the activity of human ACAT1 gene P1 promoter (DOI: 10.1038/sj.cr.7290231). Also, the authors of this manuscript have published similar upregulated expression of the ACAT-1 gene in type 2 diabetec human and a db/db mice model (doi.org/10.1155/2018/8458304). In that publication the authors have speculated on the possible role of the hyperglycemia on the expression of CD36 via activation of NF kappa B pathway mediated by proinflammatory signaling after accumulation of reactive oxygen species. Additionally, the authors have published data on usage of a Sodium-Glucose Cotransporter 2 Inhibitor to normalize the upregulated Lox-1, CD36, and ACAT-1 in db/db diabetic mice (doi.org/10.1371/journal.pone.0143396).

I have one difficult task for reviewing the current manuscript considering the previous published paper by this group. The dilemma Is to understand the significance of the current work and additive value of the presented data on the known effect of the teneligliptin on the expression of effector genes CD36 and ACAT-1 which was published earlier.

It was shown previously that ox-LDL accumulates in mouse macrophages which can be reduced by teneligliptin in diabetic db/db mice. ox-LDL-induced cholesterol ester accumulation indicated the extent of foam cell formation which was reduced by the effect of the drug. It was also shown that the reduction of foam cell formation by the drug was mediated via down regulation of the CD36 and ACAT‑1 gene expression (the actual protein expression was not shown, however). Therefore, all the evidence surrounding the central dogma of preventing atherosclerosis by drug used was achieved by the previous publication.

In this publication type 1 diabetes mice model using the effect of streptozotocin and people with type 1 diabetes were examined for the same effect. The outline of the manuscript follows the same pattern and presents similar data as type 2 diabetes. Now the question is that if the authors reason for the existence of the same pathways in type 1 diabetes as type 2 diabetes which was already elucidated it only introduces two new models and the mechanism of the action of the drug was concluded to be the same as published before.

Showing the similarity of the two types of diabetes is good but it seems that the message seems to be too short to be granted as a full publication to me.

Therefore, I suggest that the new message is expanded to the mechanisms of the action of the drug by comparing different models as a new approach to the data in hand. Also supplementing the findings with the protein expression findings as well as the data regarding IL6 in case of T1D models and patients can add value to the manuscript.

Author Response

Response to the Reviewers’ comments

Reviewer #2

The literature around the issue has evolved and part of the evidence has come from the research by the authors of this manuscript. ACAT-1 was considered as one of the key elements in atherosclerosis where it was shown for example upregulation of ACAT-1 in dexamethasone long term users possibly by enhancing the activity of human ACAT1 gene P1 promoter (DOI: 10.1038/sj.cr.7290231). Also, the authors of this manuscript have published similar upregulated expression of the ACAT-1 gene in type 2 diabetec human and a db/db mice model (doi.org/10.1155/2018/8458304). In that publication the authors have speculated on the possible role of the hyperglycemia on the expression of CD36 via activation of NF kappa B pathway mediated by proinflammatory signaling after accumulation of reactive oxygen species. Additionally, the authors have published data on usage of a Sodium-Glucose Cotransporter 2 Inhibitor to normalize the upregulated Lox-1, CD36, and ACAT-1 in db/db diabetic mice (doi.org/10.1371/journal.pone.0143396).

I have one difficult task for reviewing the current manuscript considering the previous published paper by this group. The dilemma Is to understand the significance of the current work and additive value of the presented data on the known effect of the teneligliptin on the expression of effector genes CD36 and ACAT-1 which was published earlier.

It was shown previously that ox-LDL accumulates in mouse macrophages which can be reduced by teneligliptin in diabetic db/db mice. ox-LDL-induced cholesterol ester accumulation indicated the extent of foam cell formation which was reduced by the effect of the drug. It was also shown that the reduction of foam cell formation by the drug was mediated via down regulation of the CD36 and ACAT‑1 gene expression (the actual protein expression was not shown, however). Therefore, all the evidence surrounding the central dogma of preventing atherosclerosis by drug used was achieved by the previous publication.

In this publication type 1 diabetes mice model using the effect of streptozotocin and people with type 1 diabetes were examined for the same effect. The outline of the manuscript follows the same pattern and presents similar data as type 2 diabetes. Now the question is that if the authors reason for the existence of the same pathways in type 1 diabetes as type 2 diabetes which was already elucidated it only introduces two new models and the mechanism of the action of the drug was concluded to be the same as published before.

Showing the similarity of the two types of diabetes is good but it seems that the message seems to be too short to be granted as a full publication to me.

Therefore, I suggest that the new message is expanded to the mechanisms of the action of the drug by comparing different models as a new approach to the data in hand. Also supplementing the findings with the protein expression findings as well as the data regarding IL6 in case of T1D models and patients can add value to the manuscript.

Thank you for your comments. As you pointed out, we have already reported that teneligliptin significantly reduced ox-LDL uptake, foam cell formation, CD36 and ACAT-1 mRNA levels of macrophages derived from type 2 diabetes patients and db/db mice, an animal model of type 2 diabetes. However, it remains unclear how teneligliptin exerted these beneficial effects on macrophages. In other words, the mechanisms of action of the drug are not fully elucidated. Therefore, we examined here the effects of teneligliptin on ox-LDL uptake, CD36/ACAT-1 mRNA levels in AGE-exposed mouse peritoneal macrophages and THP-1 cells, a human macrophage cell line because (1) we have previously shown that DPP-4 inhibitors could block the harmful effects of AGE in cultured endothelial cells and renal proximal tubular cells [references 24,40] and (2) AGEs play a central role in atherosclerosis in diabetes [reference 18]. In this study, we found that AGE mimicked the effects of T1D; AGE significantly increased ox-LDL uptake, CD36 and ACAT-1 gene expression in mouse macrophages and THP-1 cells, all of which were prevented by the treatment with 10 nmol/L teneligliptin. Therefore, teneligliptin could suppress the ox-LDL uptake and foam cell formation of macrophages isolated from T1D mice and patients partly via suppression of deleterious effects of AGE on macrophages. Furthermore, as you pointed out, it would be interesting to explore other genes, such as interleukin-6 with changing expression levels by teneligliptin through RNA-seq. We added these statements in the third and last paragraphs of Discussion section of revised manuscript.

Thank you again for your kindness.

Sincerely yours,

 Michishige Terasaki, M.D., Ph.D.

Division of Diabetes, Metabolism, and Endocrinology

Department of Medicine

Showa University School of Medicine

Tokyo 142-8666, Japan

Tel: +81-3-3784-8000

Fax: +81-3-3784-8948

E-mail: ttmichishige@yahoo.co.jp

Round 2

Reviewer 1 Report

Thank you to Prof Terasaki for his polite response to my review. The authors have amended the table, figures and references in line with my suggestions. Most other concerns however, have been integrated as discussion points or limitations within the Discussion section. In doing so, the actual body of work hasn't changed.

Author Response

Many thanks for your email of June 22, 2020 with your and reviewers’ valuable comments concerning our manuscripts entitled “A dipeptidyl peptidase-4 inhibitor inhibits foam cell formation of macrophages in type 1 diabetes via suppression of CD36 and ACAT-1 expression”. We are very grateful to you for your prompt arrangement and thorough review.

According to your and reviewers’ comments, the manuscript has been completely re-revised and we addressed all of them. All changes were highlighted in the manuscript by using the “Track Changes” as you requested.

We hope the re-revised version meets with your satisfaction and will be followed by publication in the special issue entitled "Immunopathology of Atherosclerosis and Related Diseases: Focus on Molecular Biology" of International Journal of Molecular Sciences.

Response to the Reviewers’ Comments

Terasaki M et al., “A dipeptidyl peptidase-4 inhibitor inhibits foam cell formation of macrophages in type 1 diabetes via suppression of CD36 and ACAT-1 expression”; Manuscript ID: ijms-816165

The authors appreciate your careful reading and constructive comments. Our point-by-point responses are provided below.

Response to the Reviewers’ comments

Reviewer #1

Thank you to Prof Terasaki for his polite response to my review. The authors have amended the table, figures and references in line with my suggestions. Most other concerns however, have been integrated as discussion points or limitations within the Discussion section. In doing so, the actual body of work hasn't changed.

Thank you for your constructive comments on our paper.

We added the following statements for clarifying the aim of the present study in the introduction section. “Recently, DPP-4 inhibitors have been shown to play a protective role against atherosclerosis in diabetic animal models [17,19] and T2D patients [20–22]. Indeed, we have found that foam cell formation and atherosclerotic lesions of diabetic apolipoprotein E-deficient mice were significantly increased compared with non-diabetic counterparts, which were prevented by vildagliptin [17]. Furthermore, an inhibitor of DPP-4, teneligliptin has been shown to significantly attenuate ox-LDL uptake and foam cell formation of macrophages isolated from obese T2D mice and patients partly by suppressing CD36 and ACAT-1 expression [14]” in line 55-61.

Thank you again for your kindness.

Sincerely yours,

                           Michishige Terasaki, M.D., Ph.D.

Division of Diabetes, Metabolism, and Endocrinology

Department of Medicine

Showa University School of Medicine

Tokyo 142-8666, Japan

Reviewer 2 Report

Many thanks for the explanations and the revision of the manuscript.

The idea of ruling out the effect of dyslipidemia, hypertension, and insulin resistance the authors have suggested to use type 1 diabetes mice models and patients to see the effect of the DPP-4 inhibitor is independent of these comorbidities. The mice models used in this study and the previous published T2D mice model should differ in the comorbidities mentioned apart from insulin resistance. Do you have data to support that the lipid panels and blood pressures are different in two mice models? It is evident that you are not studying the effect of those comorbidities in mice models on atherosclerosis but such data in relevant models should also referred to in order to support this idea. Also, the patients should differ in the same respect. Comparing both the published data and the current manuscript, the data does not support this idea since the blood pressure and lipid profiles presented in case of T1D and T2D patients and mice models do not significantly differ. Therefore, the data does not support the idea of comorbidities being a factor in this process. Therefore, the current manuscript does not show the same idea that the authors try to establish.

Also, in this manuscript the mean ages of the cases and controls differ by almost 16 years and I think the matching cannot be established for the age as the cases are generally older than the controls. Also, the sex is not matched completely since the number of male/females in case/controls do not match. This can be discussed and possible effects on the results could be also elucidated.

However, the issue of the insulin resistance is still valid in this setting. It appears to me that one possible control for the effect of T1D in mice model could be the effect of hyperglycemia on the macrophages derived from non-diabetic mice to avoid the effect of streptozotocin and narrow it down to the specific effect of the hyperglycemia on the formation of foam cells.

Additionally, the use of statins and hypertension drugs in T1D patients might also play some role in the reported results as acknowledged.

The finding of the effect of the AGEs on the expression of CD36 and ACAT-1 has already been reported by Lei Xu, 2016 (doi: 10.1186/s12944-016-0334-0) in THP-1 cells. I directly quote their results section here:

“Results: AGEs increased lipids accumulation in macrophages in a concentration-dependent manner. 600 μg/ml AGEs obviously upregulated oxLDL uptake, increased levels of cholesterol ester in macrophages, and decreased the HDL-mediated cholesterol efflux by regulating the main molecular expression including CD36, Scavenger receptors (SR) A2, HMG-CoA reductase (HMGCR), ACAT1 and ATP-binding cassette transporter G1 (ABCG1). The changes above were inversed when the cells were pretreated with anti-RAGE antibody.”

Then Lei Xu, 2016 have concluded that:

“The current study suggest that AGEs can increase lipids accumulation in macrophages by regulating cholesterol uptake, esterification and efflux mainly through binding with RAGE, which provide a deep understanding of mechanisms how AGEs accelerating diabetic atherogenesis.”

It seems in this case that AGEs can upregulate CD36 and ACAT-1 is independent of hyperglycemia which has been already shown. Now what could have been a new line of evidence would have been if the hyperglycemia in cell cultured THP-1 or macrophages can also upregulate CD36 or ACAT-1 which is missing in here.

On the other hand, the effect of the AGEs in mice model (C57/BL6) treated for 2 weeks with AGE-BSA showed impaired glucose tolerance and decrease acute insulin release in previous studies shown by Hongsoo Jung et al. (DOI: 10.1002/dmrr.1208). Therefore, it is difficult to conclude if the presence of hyperglycemia after the onset of diabetes would increase the AGEs or vice versa. For that reason the effect of the AGEs is not also novel here.

I still have difficulty seeing the data as novel in here and feel that significantly new findings needs to be presented here in order to publish the current data.

Author Response

June 27, 2020

Many thanks for your email of June 22, 2020 with your and reviewers’ valuable comments concerning our manuscripts entitled “A dipeptidyl peptidase-4 inhibitor inhibits foam cell formation of macrophages in type 1 diabetes via suppression of CD36 and ACAT-1 expression”. We are very grateful to you for your prompt arrangement and thorough review.

According to your and reviewers’ comments, the manuscript has been completely re-revised and we addressed all of them. All changes were highlighted in the manuscript by using the “Track Changes” as you requested.

We hope the re-revised version meets with your satisfaction and will be followed by publication in the special issue entitled "Immunopathology of Atherosclerosis and Related Diseases: Focus on Molecular Biology" of International Journal of Molecular Sciences.

Response to the Reviewers’ Comments

Terasaki M et al., “A dipeptidyl peptidase-4 inhibitor inhibits foam cell formation of macrophages in type 1 diabetes via suppression of CD36 and ACAT-1 expression”; Manuscript ID: ijms-816165

The authors appreciate your careful reading and constructive comments. Our point-by-point responses are provided below.

Reviewer comments

Reviewer #2

Many thanks for the explanations and the revision of the manuscript.

The idea of ruling out the effect of dyslipidemia, hypertension, and insulin resistance the authors have suggested to use type 1 diabetes mice models and patients to see the effect of the DPP-4 inhibitor is independent of these comorbidities. The mice models used in this study and the previous published T2D mice model should differ in the comorbidities mentioned apart from insulin resistance.

Do you have data to support that the lipid panels and blood pressures are different in two mice models? It is evident that you are not studying the effect of those comorbidities in mice models on atherosclerosis but such data in relevant models should also referred to in order to support this idea.

Also, the patients should differ in the same respect. Comparing both the published data and the current manuscript, the data does not support this idea since the blood pressure and lipid profiles presented in case of T1D and T2D patients and mice models do not significantly differ. Therefore, the data does not support the idea of comorbidities being a factor in this process. Therefore, the current manuscript does not show the same idea that the authors try to establish.

Thank you for your first comments. According to your comments, in order to address whether the effects of DPP-4 inhibitor are partly independent of the comorbidities associated with T2D and pleiotropic actions of incretins, we compared body weight, blood pressure, lipid and glycemic parameters, including homeostasis model assessment of insulin resistance (HOMA-IR), a marker of insulin resistance of T1D mice and patients with those of T2D.

For your evaluation: Table 1

T2D model db/db mice

T1D model mice

p-value

Age

13-weel old

13-week old

Number

7

6

Final body weight (g)

43.8 ± 1.5

20.3 ± 3.2

P < 0.001

Food Intake (g/day)

4.5 ± 0.5

4.8 ± 1.0

0.43

SBP (mmHg)

109 ± 10

102 ± 15

0.302

DBP (mmHg)

61 ± 5

65 ± 7

0.296

Total-C (mg/dL)

112 ± 42

86 ± 22

0.202

HDL-C (mg/dL)

63 ± 33

32 ± 15

0.06

Triglycerides (mg/dL)

134 ± 42

82 ± 39

p < 0.05

FBG (mg/dL)

432 ± 68

164 ± 56

p < 0.001

Insulin (ng/mL)

2.48 ± 0.85

0.05 ± 0.02

p < 0.001

HOMA-IR

71 ± 32

0.53 ± 0.27

p < 0.001

HbA1c (%)

8.3 ± 0.6

7.9 ± 0.5

0.2125

OGTT-AUC of glucose

(mg/dL x hour)

1215 ± 225

1150 ± 365

0.6997

For your evaluation: Table 2

T2D patients

T1D patients

p-value

Number (male/female)

4 (4/0)

5 (3/2)

0.152

Age (years)

63 ± 18

58 ± 30

0.754

Duration of diabetes (years)

19 ± 12

11 ± 10

0.322

Body Weight (kg)

74 ± 19

71 ± 20

0.842

BMI (kg/m2)

26.4 ± 6.4

21.3 ± 3.9

0.178

SBP (mmHg)

113 ± 5

114 ± 10

0.892

DBP (mmHg)

70 ± 6

68 ± 13

0.845

Total-C (mg/dL)

187 ± 20

167 ± 24

0.224

LDL-C (mg/dL)

119 ± 17

92 ± 28

0.144

HDL-C (mg/dL)

39 ± 9

57 ± 20

0.146

Triglycerides (mg/dL)

149 ± 16

90 ± 48

p < 0.05

FBG (mg/dL)

189 ± 48

233 ± 78

0.354

HbA1c (%)

8.3 ± 0.5

7.7 ± 0.4

0.081

Fasting C-peptide (ng/mL)

1.5 ± 0.7

0.16 ± 0.09

p < 0.005

Retinopathy (NDR/SDR/PPDR/PDR)

(2/0/0/2)

(4/1/0/0)

0.165

Nephropathy (1/2/3/4/5)

(0/3/1/0/0)

(3/1/1/0/0)

0.140

PAD/none

(2/2)

(1/4)

0.343

Total daily insulin dose (Unit)

48 ± 24

48 ± 36

0.993

Lipid-lowering drugs (statins/none)

(4/0)

(2/3)

0.058

Anti-hypertensive drugs

(ARBs/ARBs+CCBs/CCBs/none)

(3/0/0/1)

(1/1/1/2)

0.353

As shown in the Table 1 and 2 for your evaluation, body weight, triglycerides and homeostasis model assessment of insulin resistance (HOMA-IR), an index of insulin resistance were higher in the previously published T2D mice models or patients compared with controls, and there were significant differences of triglycerides and HOMA-IR between T1D and T2D. We added the following sentences in the first paragraph of Discussion section in the revised manuscript in line 156-169: Therefore, it remains unclear whether DPP-4 inhibitors slow the process of atherosclerosis directly via the improvement of hyperglycemia and related AGE formation, indirectly via the amelioration of comorbid risk factors, or through the pleiotropic effects of incretins. To address whether the effects of DPP-4 inhibitor are partly independent of these comorbidities associated with T2D and pleiotropic actions of incretins, we first examined the effects of a DPP-4 inhibitor, teneligliptin on ox-LDL uptake and foam cell formation of macrophages isolated from STZ-induced T1D mice and T1D patients because (1) blood pressure levels and lipid parameters in T1D mice and patients were comparable with those in controls, (2) body weight, triglycerides, and homeostasis model assessment of insulin resistance (HOMA-IR), a marker of insulin resistance were higher in the previously published T2D mice models or patients than controls, (3) there were significant differences of triglycerides and HOMA-IR or fasting C-peptide between T1D and T2D, and (4) DPP-4 inhibitors did not modulate the effects of incretins on macrophages in vitro. In other words, we used here macrophages from T1D mice models and patients to rule out the effects of these comorbidities.

Further, we added the following statements for more clarifying the aim of the present study in the introduction sections: “Recently, DPP-4 inhibitors have been shown to play a protective role against atherosclerosis in diabetic animal models [17,19] and T2D patients [20–22]. Indeed, we have found that foam cell formation and atherosclerotic lesions of diabetic apolipoprotein E-deficient mice were significantly increased compared with non-diabetic counterparts, which were prevented by vildagliptin [17]. Furthermore, an inhibitor of DPP-4, teneligliptin has been shown to significantly attenuate ox-LDL uptake and foam cell formation of macrophages isolated from obese T2D mice and patients partly by suppressing CD36 and ACAT-1 expression [14]” in line 55-61.

Also, in this manuscript the mean ages of the cases and controls differ by almost 16 years and I think the matching cannot be established for the age as the cases are generally older than the controls. Also, the sex is not matched completely since the number of male/females in case/controls do not match. This can be discussed and possible effects on the results could be also elucidated.

Thank you for your second comments. We totally agreed with your opinion. Indeed, age and sex were not completely matched between T1D cases and controls. Since accumulation of AGE has been known to progress in a normal ageing process and under diabetic conditions, especially diabetic patients with a long disease history [45], the difference of mean ages of the T1D cases and controls may also influence the present results. In addition, there is a sex disparity in cardiovascular event and mortality rates associated with diabetes [Yamagishi SI. Diabetes Metab Res Rev. 2018: e3059], and therefore the difference of number of male/female in cases/controls may also affect the ox-LDL uptake and foam cell formation of macrophages. We added these sentences in the last paragraph of Discussion section in the revised manuscript in line 209-215. We also cited a new reference [47].

However, the issue of the insulin resistance is still valid in this setting. It appears to me that one possible control for the effect of T1D in mice model could be the effect of hyperglycemia on the macrophages derived from non-diabetic mice to avoid the effect of streptozotocin and narrow it down to the specific effect of the hyperglycemia on the formation of foam cells.

Thank you for your third comments. As to your comments, in order to avoid the effects of streptozotocin and narrow it down the specific effect of hyperglycemia on macrophages, we investigated the effects of AGE, a marker of cumulative hyperglycemic exposure on ox-LDL uptake, CD36 and ACAT-1 gene expression of macrophages derived from non-diabetic mice (Figure 2A-D, Figure 2I and J). We added these sentences in the last paragraph of Discussion section in the revised manuscript in line 216-219.

Additionally, the use of statins and hypertension drugs in T1D patients might also play some role in the reported results as acknowledged.

Thank you for your fourth comment. We totally agreed with your opinion. As you pointed out, it is probable that any of clinical characteristics of T1D patients, including disease duration, age and drug medications, such as statins and anti-hypertensive agents could have impacted the present findings. We added these sentences in the last paragraph of Discussion section in the revised manuscript in line 206-208.

The finding of the effect of the AGEs on the expression of CD36 and ACAT-1 has already been reported by Lei Xu, 2016 (doi: 10.1186/s12944-016-0334-0) in THP-1 cells. I directly quote their results section here:

“Results: AGEs increased lipids accumulation in macrophages in a concentration-dependent manner. 600 μg/ml AGEs obviously upregulated oxLDL uptake, increased levels of cholesterol ester in macrophages, and decreased the HDL-mediated cholesterol efflux by regulating the main molecular expression including CD36, Scavenger receptors (SR) A2, HMG-CoA reductase (HMGCR), ACAT1 and ATP-binding cassette transporter G1 (ABCG1). The changes above were inversed when the cells were pretreated with anti-RAGE antibody.” Then Lei Xu, 2016 have concluded that: “The current study suggest that AGEs can increase lipids accumulation in macrophages by regulating cholesterol uptake, esterification and efflux mainly through binding with RAGE, which provide a deep understanding of mechanisms how AGEs accelerating diabetic atherogenesis.”

It seems in this case that AGEs can upregulate CD36 and ACAT-1 is independent of hyperglycemia which has been already shown. Now what could have been a new line of evidence would have been if the hyperglycemia in cell cultured THP-1 or macrophages can also upregulate CD36 or ACAT-1 which is missing in here.

Thank you for your fifth comments. As you mentioned, Xu L, et al. already reported that AGE at supraphysiological concentrations (300-600 µg/mL AGE-BSA) increased lipid accumulation in macrophages partly by regulating CD36, scavenger receptor A2, hydroxymethylglutaryl-CoA reductase, ACAT-1 and ATP-binding cassette transporter G1 [Xu L, et al. Lipids Health Dis. 2016, 15, 161.]. However, we provided here a new line of following evidence; concentration of AGE (100 µg/mL AGE-BSA), which is comparable with that of in vivo-diabetic situation [Yamagishi SI, et al, Mol Med 2015, 21, S32-40. Takeuchi M, et al., Mol Med 1999, 5, 393-405.] acutually stimulated ox-LDL uptake of macrophages via CD36 and ACAT-1 expression, which is a molecular target for atheroprotective properties of teneligliptin. We added these statements in the last paragraph of Discussion section in line 230-236. We cited three references [48], [49] and [50] in the revised manuscript.

On the other hand, the effect of the AGEs in mice model (C57/BL6) treated for 2 weeks with AGE-BSA showed impaired glucose tolerance and decrease acute insulin release in previous studies shown by Hongsoo Jung et al. (DOI: 10.1002/dmrr.1208). Therefore, it is difficult to conclude if the presence of hyperglycemia after the onset of diabetes would increase the AGEs or vice versa. For that reason the effect of the AGEs is not also novel here.

Thank you for your sixth comments. As you pointed out, intraperitoneal administration of AGE-BSA has been reported to impair glucose tolerance in mice in association with decrease in acute insulin secretion [DOI: 10.1002/dmrr.1208]. AGE may augment foam cell formation by further deteriorating hyperglycemia. We added these statements in the last paragraph of Discussion section in the revised manuscript in line 236-238. We cited a reference [51] in the revised manuscript.

I still have difficulty seeing the data as novel in here and feel that significantly new findings needs to be presented here in order to publish the current data.

Thank you for your last comments. As to your comments, the salient findings of our present study is that teneligliptin at a therapeutic level inhibited foam cell formation of macrophages by suppressing the harmful effects of AGE, whose concentration is also comparable with that of diabetic condition. We added these sentences in the last paragraph of Discussion section in the revised manuscript in line 238-241.

Thank you again for your kindness.

Sincerely yours,

                           Michishige Terasaki, M.D., Ph.D.

Division of Diabetes, Metabolism, and Endocrinology

Department of Medicine

Showa University School of Medicine

Tokyo 142-8666, Japan

Tel: +81-3-3784-8000

Fax: +81-3-3784-8948

E-mail: ttmichishige@yahoo.co.jp

Round 3

Reviewer 1 Report

I have no further comments or suggestions

Reviewer 2 Report

Dear Editor,

I shall thank you for the opportunity to see this version of the manuscript.

It seems that all the comments have beed discussed in the manuscript and I do not have further comments at this stage.

With the best regards